# Real-time forecasting of data revisions in epidemic surveillance streams

Jingjing Tang[1]*, Aaron Rumack[2], Bryan Wilder[2], Roni Rosenfeld[2]

**1** Computational Biology Department, Carnegie Mellon University, Pittsburgh, Pennsylvania, United States of America, **2** Machine Learning Department, Carnegie Mellon University, Pittsburgh, Pennsylvania, United States of America

\* jtang2@andrew.cmu.edu

## Abstract

Epidemic data streams undergo frequent revisions due to reporting delays ("backfill") and other factors. Relying on tentative surveillance values can seriously degrade the quality of situational awareness, forecasting accuracy and decision-making. We introduce Delphi Revision Forecast (Delphi-RF), a real-time data revision forecasting framework using nonparametric quantile regression, applicable to both counts and proportions (fractions) in public health reporting. By incorporating all available revisions up to a given estimation date, Delphi-RF models revision dynamics and generates distributional forecasts of finalized surveillance values. Applied to daily COVID-19 data (insurance claims, antigen tests, confirmed cases) and weekly dengue and influenza-like illness (ILI) case counts, Delphi-RF delivers accurate revision forecasts, particularly in early reporting stages. In addition, it improves computational efficiency by more than 10-100x compared to existing methods, making it a scalable solution for real-time public health surveillance.

## Author summary

Accurate and reliable forecasts of infectious disease epidemics, such as COVID-19, are essential but challenging. The presence of data revisions in public health data streams can introduce significant biases in both predictors and responses, leading to suboptimal situational awareness, preparedness, and downstream countermeasure design. To address this issue, we propose a modeling framework that leverages historical revision patterns to generate distributional forecasts of finalized surveillance values. Applicable to both count-type and fraction-type data across various temporal resolutions and epidemic surveillance data streams, our approach ensures real-time accuracy, even with only early revisions available. Moreover, our method achieves competitive or superior forecast accuracy compared to existing methods, while also demonstrating a more than 10-100x improvement in computational efficiency.

**Data availability statement:** The training data used in this study are publicly available from the Delphi Epidata API at: https://cmu-delphi.github.io/delphi-epidata/.

The Massachusetts Department of Public Health provides a comprehensive revision history of COVID-19 case reports at: https://www.mass.gov/info-details/archive-of-covid-19-cases-in-massachusetts. All code and preprocessed data required to reproduce the results and run the models are available at: https://github.com/cmu-delphi/data-revisions-forecasting-paper.

**Funding:** JT was supported by a fellowship from the Center for Machine Learning and Health at Carnegie Mellon University (https://www.cs.cmu.edu/cmlh-cfp). JT, BW and RR were supported by Centers for Disease Control and Prevention grant under award No.U01IP001121 and contract No. 75D30123C1590. (https://www.cdc.gov). The funders had no role in study design, data collection and analysis, decision to publish, or preparation of the manuscript.

## Introduction

The COVID-19 pandemic, as a global public health crisis, has precipitated unprecedented societal, economic, and political disruptions, underscoring the imperative of real-time epidemic forecasting. However, many of the epidemic surveillance values published by public health surveillance data systems are often and repeatedly revised in subsequent releases after their initial release, and do not accurately reflect disease activity in real time. This often leads to biased and error-prone situational awareness [1,2] and poses substantial hurdles to achieving real-time epidemic forecast accuracy.

The impact of reporting delays and subsequent data revisions has been discussed not only in the context of public health studies: from influenza [3] to dengue [4] to COVID-19 forecasting [5,6], but also in the macroeconomic domain [7]. Data revisions arise from various factors, including error corrections, infrastructure limitations, and varying delays between data collection and reporting [3,8]. These factors affect surveillance values differently, leading to distinct data revision patterns. For example, case counts typically increase monotonically during the revision process, a phenomenon commonly referred to as "backfill". However, epidemic fractions (e.g., the percentage of positive COVID-19 insurance claims out of total claims) can fluctuate dramatically—either increasing or decreasing—because the numerator and denominator often exhibit different backfill dynamics.

Several studies have addressed data revision and reporting delay issues, particularly in the context of seasonal infectious diseases with extensive weekly surveillance data. Early approaches relied on relatively simple statistical models. For instance, linear regression was used to adjust provisional data and forecast ILI case counts across 15 Latin American countries [9], while the residual density method was applied to estimate the distribution of revised updates in weekly ILI data [10]. More recent efforts have focused on probabilistic and Bayesian frameworks to better handle uncertainty and temporal variation in delays. A flexible Bayesian model, Nowcasting by Bayesian Smoothing (NobBS), was introduced to accommodate time-varying delay distributions and improve uncertainty quantification for dengue and ILI case counts [1]. Generalized Bayesian methods using Laplace approximation were applied to dengue and SARI data in Brazil [11], and a generalized Dirichlet-multinomial mixture model was proposed for weekly dengue data in Rio de Janeiro [12]. Other approaches have incorporated structured or semi-mechanistic models. For example, to nowcast delayed norovirus cases in England during winter 2023/24, three models—a generalized additive model, a Bayesian structural time series model incorporating syndromic surveillance data, and the semi-mechanistic Bayesian delay model EpiNowcast [13,14] - were developed and evaluated within a common probabilistic scoring framework. Some models also account for backfill uncertainty without explicitly modeling its dynamics [15].

Compared with seasonal infectious diseases, COVID-19 surveillance data are noisier and less regular, making it harder to extract stable features for revision modeling. To address these challenges, several recent methods have been proposed. A neural network-based framework has been proposed to refine COVID-19 forecasts

using weekly case counts [16]. Although effective, this method requires substantial computational resources and does not account for the statistical properties inherent in public health datasets. Besides, a Bayesian spatiotemporal nowcasting model was introduced to estimate COVID-19 case counts at the county level in Ohio, incorporating an autoregressive structure to capture temporal dynamics [17]. Both methods focus on count-type data and were evaluated only during the pre-Delta phase of the pandemic.

Among existing methods, Epinowcast [14] stands out as a strong competitor leveraging a full Bayesian framework for nowcasting with robust uncertainty quantification. However, its reliance on Bayesian inference makes it computationally intensive, often requiring long runtimes that can limit real-time applicability. Moreover, EpiNowcast and similar Bayesian models are designed primarily for count-type data and are less adaptable to fraction-type quantities. This poses an important limitation, since many key public health indicators—such as antigen test positivity rates, hospitalization ratios, and syndromic surveillance measures—are expressed as fractions rather than counts. The difficulty arises because Bayesian methods typically assume an underlying mechanistic process governing the evolution of counts. While mechanistic models for fractions can, in principle, be specified, in practice such revision data often reflect administrative or behavioral processes that are difficult to represent mechanistically.

In the broader literature, data revisions and real-time analysis have been extensively studied in macroeconomics. Notably, comprehensive surveys have been conducted on these topics [18,19], and various modeling approaches—such as state-space models—have been summarized [7]. However, these macroeconomic methods are not directly transferable to public health contexts. Revisions in public health data are driven by health-seeking behavior and the administrative practices of public health agencies which are influenced by operational constraints, staffing capacity, and evolving reporting protocols, etc.

In this paper, we introduce Delphi Revision Forecast (Delphi-RF), a robust and operational framework for correcting data revisions, which is openly available at https://github.com/cmu-delphi/DelphiRF. The system leverages historical revisions to estimate the probability distribution of finalized values, which become fully available only at a later stage (Fig 1). We focus on the distributional forecasting of finalized surveillance values in *real time*. For clarity of terminology, we use the term *data revision forecasting* to refer to the problem of predicting current quantities based solely on their preliminary measurements. We reserve the term *nowcasting* to refer to the more general problem of predicting current quantities based on any data available currently, including other data sources. Delphi-RF is designed to accommodate signals with varying temporal resolutions and directly models the revision process through flexible regression features, without assuming an explicit generative structure. This makes it naturally applicable to both count- and fraction-type targets.

The Delphi Group at Carnegie Mellon University curates real time infectious disease indicators and makes them accessible via public API [20–22]. When applied to a variety of such indicators, we have shown that Delphi-RF provides competitive or superior forecast accuracy for data revisions, including for COVID-19, dengue fever, and ILI. Moreover, Delphi-RF achieves an over 10- to 100-fold improvement in computational efficiency compared to existing methods, making it a scalable and practical solution for real-time public health surveillance.

The remainder of this paper is structured as follows. Terminology, notation and problem definition introduces the problem formulation, along with key terminology and notation. Methods presents the proposed model, the evaluation framework, and the adaptive modeling protocol. Experimental results describes the datasets, preprocessing steps, and experimental setup, followed by results that demonstrate the model's performance across multiple COVID-19 indicators, along with comparative analyses using alternative methods for other infectious diseases, an ablation study, and a hyper-parameter sensitivity analysis. Finally, Conclusion and discussion concludes with a discussion of the findings and outlines directions for future research.

## Terminology, notation and problem definition

Throughout this paper, we use the term reference date to refer to the date $t$ associated with a particular quantity, and report date to refer to the date $s$ on which that quantity becomes available.

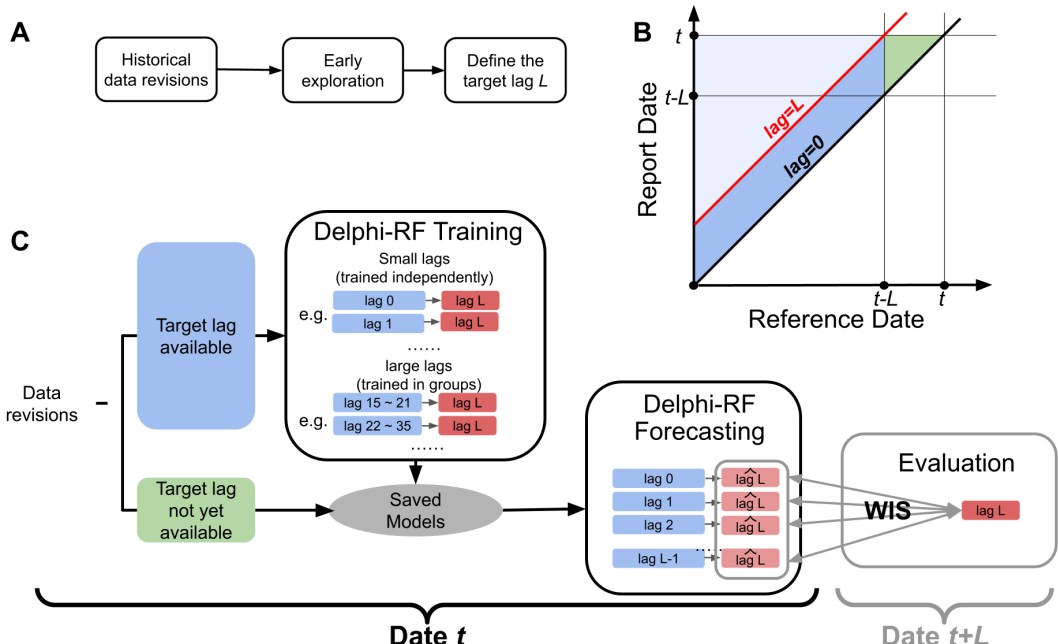

**Fig 1. Overview of the Delphi-RF framework.** (A) Preparatory step: early exploration of historical revisions to define the target lag $L$ before model training and forecasting. (B) Data structure in the report date–reference date space. Data above the lag = 0 diagonal correspond to revisions that are available (or will become available). The light blue region shows revisions with lag greater than the target lag, which are excluded from Delphi-RF. The blue parallelogram marks revisions with the target lag available, which are used for training. The green triangular region indicates the most recent revisions, where the target lag is not yet available. (C) Workflow of Delphi-RF: revisions with target lag available are used for training (small lags trained independently, large lags trained in groups). Saved models are then applied to real-time revisions for forecasting. $L$ days later, predictions are evaluated using the Weighted Interval Score (WIS).

Let $Y_{it}$ denote a surveillance value associated with the *reference date t* for location $i$. The time series $\{Y_{it}\}_{t \in T}$ represents the usual uni-variate surveillance data for location $i$, where $T$ denotes a set of reference dates. The value of $Y_{it}$ reported as of date $s$ ($s \geq t$) is denoted by $Y_{itl}$, where $l$ is the *lag*, defined as $l = s - t \in \mathbf{N}$, representing the number of days between the report date $s$ and the reference date $t$. By convention, $l \geq 0$.

**Revision dynamics:** Due to the existence of data revision, each $Y_{it}$ has its *data revision sequence* $Y_{it}^{t:s}$ as of date $s$, equivalently denoted by $Y_{it,0:(s-t)}$. Only a minimal portion of data revision results from instances where a case is initially lost but subsequently recovered, or when a case is initially entered incorrectly but corrected later. Most data revisions are typically the result of reporting delays. The data revision sequence $Y_{it,0:l}$ tends to asymptote as $l$ approaches a sufficiently large value $L_{it}$.

By definition, $Y_{itl}$ represents the most recent (up-to-date) version of $Y_{it}$ as of date $s = t + l$ regardless of whether an explicit report or revision occurred on date $s$. In this convention, no reported revision is treated as equivalent to a report of no revision, since these two cannot usually be distinguished in the available data. Moreover, if $Y_{it}$ has never been reported until lag $l$, then all prior lags are treated as reports of zero. Formally, we set

$$Y_{itl} = \begin{cases} 0, & \text{if } l = l_{\min} \\ Y_{it(l-1)}, & \text{if } l \geq l_{\min} + 1 \end{cases}$$

where $l_{\min}$ denotes the smallest reporting lag at which the dataset provides its initial report. Unlike the idealized setting where data are assumed to be available at lag 0 (same-day reporting), in practice data sources can release their initial reports only after a positive delay.

**Problem setup:** In practice, the time required for the convergence of the revision exhibits considerable variability across different data streams, locations $i$ and reference dates $t$, and can be exceptionally large. S1 Fig provides an illustration in which, for COVID-19 claims reports with a reference date of 2021-08-01, most states require more than 180 days (approximately half a year) for the data revision sequences to converge.

However, such values of $L_{it}$ are not available in real-time since it is impossible to determine whether the revision for $Y_{it}$ has been finalized. This creates a challenge in selecting the target value corresponding to a target horizon $L_{it}$ for $Y_{it}$. On one hand, we prefer a long target horizon to ensure that the reporting sequence has asymptoted or is close to asymptoting. We aim for greater accuracy, which means that we tend to select a *larger lag $L_{it}$* that is large enough to ensure that the estimates closely approximate the value to which $Y_{it}$ asymptotes. On the other hand, we want our model to remain adaptive. Data revision dynamics evolve over time, and training on outdated data could result in model mismatch or bias. Selecting a target lag $L_{it}$ limits the model to data from $L_{it}$ days ago, whereas a smaller target lag allows the model to respond more effectively to recent changes in the data.

The selection of the target lag involves a trade-off between accuracy and adaptability. To address this, we choose a fixed target lag $L$ for all reference dates and locations that captures the majority of revisions (e.g., 90% of case counts reported) after exploring the available revision history of a public health data stream. Additionally, we ensure that the target lag does not exceed 60 days to maintain the model's adaptability.

Given a revision sequence $Y_{it,0:l}$, our objective is to produce a distributional estimate of the target value $Y_{itL}$ for a suitably large $L$, expressed as a set of estimated quantiles $Q^{\tau}_{Y_{itL}}$ corresponding to a predefined set of quantile levels $\tau$.

## Methods

In this section, we introduce a non-parametric model, Delphi Revision Forecast (Delphi-RF), designed to forecast the dynamics of data revisions. Unlike parametric approaches that assume a specific distributional form of the data revisions, Delphi-RF employs quantile regression [23–25], which provides a flexible framework for modeling the conditional distribution of revision magnitudes. This allows the model to capture heterogeneous revision behavior across time, locations, and data sources without imposing restrictive assumptions.

Let $L$ denote the target lag. For a random variable $Y_{itl}$ representing the value for location $i$ and reference date $t$, as reported at time $t + l$, the corresponding target is $Y_{itL}$. We denote the cumulative distribution function of $Y_{itL}$ as

$$F_{Y_{itL}}(y) = P(Y_{itL} \leq y).$$

The $\tau$th quantile of $Y_{itL}$ is defined as

$$Q^{\tau}_{Y_{itL}} = \inf\{y : F_{Y_{itL}}(y) \geq \tau\}, \quad \tau \in (0, 1).$$

Given the potential non-linear effects of calendar factors such as day-of-week [26,27] and week-of-month, and motivated by the objective of minimizing relative error between estimates and targets, we adopt a multiplicative model to estimate the conditional quantiles of the log-transformed target. To avoid undefined values when reported counts are zero, we apply a natural logarithmic transformation, defined as

$$f(y) = \begin{cases} \log(y + 1), & \text{if } y \text{ is a count} \\ \log(y^{\text{num}} + 1) - \log(y^{\text{denom}} + 1), & \text{if } y \text{ is a fraction} \end{cases}$$

where in the fraction setting, $y = y^{num}/y^{denom}$, with $y^{num}$ and $y^{denom}$ denoting the reported numerator (e.g., number of positive tests) and denominator (e.g., total number of tests), respectively. Since $f(\cdot)$ is a monotone increasing function in the count setting, the quantile of the transformed target equals the transformed quantile of the target:

$$Q^\tau_{f(Y_{itL})} = f\left(Q^\tau_{Y_{itL}}\right).$$

In the fraction setting, this exact equivalence does not strictly hold due to the separate transformations of numerator and denominator; however, when both components are sufficiently large, the transformation is effectively monotone in the ratio, and the equivalence holds to a close approximation in practice.

At any given estimation date (a report date) $s_0$, our goal is to make distributional estimates of $Y_{itL}$ for all reference date $t \in (s_0 - L, s_0]$ based on data that is available as of date $s_0$. To simplify notation, we use $f(Y_{itL})|X_{itl}$ to represent $f(Y_{itL})$ as conditioned on the feature vector $X_{itl}$, which is based on $\{Y_{itl}\}_{t+l \leq s_0, l \in [0,L)}$. Therefore, our model is

$$Q^\tau_{f(Y_{itL})|X_{itl}} = X_{itl}\beta^\tau$$

We incorporate features to account for week-of-month effects based on report dates, as well as day-of-week effects based on both report dates and reference dates. To capture week-of-month effects, we use the indicator $\mathbf{I}_{first\text{-}week(t)}$, which identifies whether a given date $t$ falls within the first week of a month, where each week begins on a Sunday. If date $t$ corresponds to the final days of a month and overlaps both the fifth week of the current month and the first week of the subsequent month, it is still classified as part of the first week. For day-of-week effects, we define the vector $\mathbf{e}_{wd(t)}$ as a one-hot encoded vector, where the first element is set to 0 if $t$ is a Monday, 1 if $t$ falls on a weekend, and 2 otherwise. To ensure model identifiability, we omit one category from each of the two one-hot encoded feature sets.

We incorporate two features to represent disease activity levels. The first one is the 7-day moving average of the current reports. Let $\widetilde{Y}_{itl}$ denote the 7-day moving average of values reported as of report date $t + l$, defined as

$$\widetilde{Y}_{itl} = \begin{cases} \frac{1}{7}\sum_{v=0}^{6} Y_{i(t-v)(l+v)}, & \text{in the count setting} \\ \frac{\sum_{v=0}^{6} Y^{num}_{i(t-v)(l+v)}}{\sum_{v=0}^{6} Y^{denom}_{i(t-v)(l+v)}}, & \text{in the fraction setting} \end{cases}$$

where in the fraction setting, $Y_{itl} = Y^{num}_{itl}/Y^{denom}_{itl}$, with $Y^{num}_{itl}$ and $Y^{denom}_{itl}$ denoting the reported numerator and denominator, respectively.

To capture changes in revision patterns, we introduce two extra set of features: 1) $f(\widetilde{Y}_{i(t-1)(l+1)}), f(\widetilde{Y}_{i(t-7)(l+7)})$ which are the most recent revision for the reference date 1 day and 7 days ago, which can provide extra information about how the epidemic trend changes in the near history; 2) $(f(\widetilde{Y}_{i(t-1)(l+1)}) - f(\widetilde{Y}_{i(t-1)l_{min}})), (f(\widetilde{Y}_{i(t-7)(l+7)}) - f(\widetilde{Y}_{i(t-7)l_{min}})$ how much the revision is made in the latest release for the reference date $t$–1 and $t$–7 compared to their initial release. This design, which incorporates both the exact value of the most recent revisions and the magnitude of change relative to the initial release, serves to reduce noise and improve numerical stability, as the initial releases are often small and noisy.

To capture changes in revision patterns, we introduce two additional sets of features. The first consists of the most recent revisions for the reference dates $t$–1 and $t$–7, defined as $f(\widetilde{Y}_{i(t-1)(l+1)})$ and $f(\widetilde{Y}_{i(t-7)(l+7)})$, which provide information on short-term epidemic trends. The second set measures the magnitude of revision for these same reference dates relative to their initial releases, given by $f(\widetilde{Y}_{i(t-1)(l+1)}) - f(\widetilde{Y}_{i(t-1)l_{min}})$ and $f(\widetilde{Y}_{i(t-7)(l+7)}) - f(\widetilde{Y}_{i(t-7)l_{min}})$. This design captures both the current epidemic intensity and the magnitude of revisions relative to the initial report, offering insight into how strongly early reports are updated across different levels of disease activity. It also enhances numerical stability, as initial releases are typically small and highly variable.

Now, the full model can be expressed as:

$$Q^{\tau}_{f(Y_{itL})|X_{itl}}$$
$$=X_{itl}\beta^{\tau}$$
$$=\beta^{\tau}_0 + f(\widetilde{Y}_{itl})\beta^{\tau}_1 \qquad \text{(Intercept, disease activity level)}$$
$$+ I_{\text{first-week}(t+l)}\beta^{\tau}_2 \qquad \text{(Week-of-month effect)}$$
$$+ \mathbf{e}_{wd(t)}\beta^{\tau}_{3:4} + \mathbf{e}_{wd(t+l)}\beta^{\tau}_{5:6} \qquad \text{(Day-of-week effects)}$$
$$+ \left( f(\widetilde{Y}_{i(t-1)(l+1)}) - f(\widetilde{Y}_{i(t-1)l_{\min}}) \right)\beta^{\tau}_7 \qquad \text{(Recent revision magnitude, } t-1\text{)}$$
$$+ \left( f(\widetilde{Y}_{i(t-7)(l+7)}) - f(\widetilde{Y}_{i(t-7)l_{\min}}) \right)\beta^{\tau}_8 \qquad \text{(Recent revision magnitude, } t-7\text{)}$$
$$+ f(\widetilde{Y}_{i(t-1)(l+1)})\beta^{\tau}_9 + f(\widetilde{Y}_{i(t-7)(l+7)})\beta^{\tau}_{10} \qquad \text{(Short-term epidemic trends)}$$

We estimate the coefficients by solving the following regularized quantile regression problem:

$$\beta^{\tau} = \arg\min_{\beta} \sum_{t=s_0-L-W}^{s_0-L} \sum_{l=\max(l_{\min},l-c)}^{\min(L-1,l+c)} w_{itl} \cdot \rho_{\tau}\left( f(Y_{itL}) - X_{itl}\beta \right) + \lambda\|\beta\|_1.$$

where $\rho_{\tau}(\cdot)$ denotes the quantile loss function [23], and $\| \cdot \|_1$ is the $\ell_1$-norm.

The flexibility and adaptability of this framework are governed by four key hyperparameters, each influencing a different dimension of the training procedure. These hyperparameters determine how data are selected, weighted, and regularized during model estimation:

1. **Regularization strength** ($\lambda$): An $\ell_1$ (Lasso) penalty is applied to the coefficient vector to promote sparsity in the model, thereby reducing overfitting and enhancing interpretability. The hyperparameter $\lambda$ controls the strength of this regularization and governs the trade-off between model complexity and fit.
2. **Training window length** ($W$): Instead of using the entire historical dataset, we restrict training to the most recent $W$ days for which the target is available prior to the evaluation time. This temporal constraint ensures that the model focuses on recent reporting behavior while still incorporating sufficient historical information for effective training.
3. **Lag padding** ($c$): Because data revision patterns vary substantially across reporting lags, we modify the regularized data revision correction framework by narrowing the lag window and training separate models for quantities reported at different lags. In theory, this is equivalent to fitting a single generalized linear model to the pooled dataset. However, this equivalence breaks down under $\ell_1$ regularization, as the lasso alters the solution space by favoring sparsity and reducing sensitivity to outliers.

   To estimate the quantities reported at lag $l$, we define the training set over a local neighborhood of lags, $\mathcal{L}(l, c) = \{l' : l - c \leq l' \leq l + c\}$, where $c$ controls the width of the lag window. When $c > 0$, the inverse lag feature ($1/(l + 1)$) is included to reflect lag-dependent effects across neighboring lags.

   Although this strategy requires fitting multiple models and incurs additional computational cost, it improves estimation accuracy by better capturing lag-specific revision dynamics under regularization.
4. **Decay parameter** ($\gamma$): To emphasize training examples that resemble the current epidemic context, we introduce sample-specific weights:

$$w_{itl} = \exp(-\gamma \cdot D^y_{itl} \cdot D^s_{itl}),$$

where $\gamma \geq 0$ controls the sharpness of the weighting scheme. The weight $w_{itl}$ is computed based on the product of two similarity measures, evaluated relative to the estimation date $s_0$:

- $D_{itl}^y = \left| f(\widetilde{Y}_{i(s_0-l)l}) - f(\widetilde{Y}_{itl}) \right|$ quantifies the difference in activity levels between the current observation and the most recent report at lag $l$, measured on the log scale.
- $D_{itl}^s = \left| [f(\widetilde{Y}_{i(s_0-l)l}) - f(\widetilde{Y}_{i(s_0-l-7)(l+7)})] - [f(\widetilde{Y}_{itl}) - f(\widetilde{Y}_{i(t-7)(l+7)})] \right|$ captures the difference in 7-day trends between the two time points.

Larger values of $\gamma$ place greater emphasis on samples with similar epidemic behavior, allowing the model to focus on training points most representative of current conditions.

## Evaluation metrics

We use the Weighted Interval Score (WIS) [28], a standard metric for evaluating distributional forecasts, to quantify the distance between the forecast distribution $F$ and the target variable $Y$.

$$\mathrm{WIS}(F, Y) = 2 \sum_\tau \phi_\tau(Y - Q_Y^\tau)$$

where $\phi_\tau(x) = \tau|x|$ for $x \geq 0$ and $\phi_\tau(x) = (1-\tau)|x|$ for $x < 0$, which is called the tilted absolute loss [21]. $Q_Y^\tau$ denotes the forecasted $\tau$th quantile of $Y$. Given a certain estimation task of $Y_{it}$ for location $i$ and reference date $t$ based on the quantities of interest that is available on date $t + l$, the WIS score can be written as

$$\mathrm{WIS}(F_{f(Y_{itL}|X_{itl})}, f(Y_{itL})) = 2 \sum_\tau \phi_\tau(f(Y_{itL}) - Q_{f(Y_{itL}|X_{itl})}^\tau)$$

where the set $\{Q_{f(Y_{itL})|X_{itl}}^\tau\}_\tau$ represents the forecast distribution over quantiles for the log-transformed target value $Y_{itL}$, where $Y_{itL}$ denotes the $L$th revision of $Y_{it}$. If only the median is forecasted, the WIS reduces to the absolute error on the log scale:

$$\mathrm{WIS}_{itl} = |f(Y_{itL}) - Q_{f(Y_{itL})|X_{itl}}^{0.5}|$$

Since WIS is computed on the log scale, it adopts a symmetric perspective on relative error, ensuring scale invariance and robustness to variation in magnitude across different reference dates and locations. However, when the target value approaches zero, relative errors can become highly volatile, introducing sensitivity into the evaluation metric.

The quantity $\exp(\mathrm{WIS}) - 1$ approximates the absolute percentage error (APE), allowing for an interpretable link between the log-scale WIS and relative error in the original scale. A smaller $\mathrm{WIS}_{itl}$ therefore indicates a smaller relative error between the distributional forecast and the target. When only the median forecast is considered, $\exp(\mathrm{WIS}) - 1$ coincides with the APE if the projected median is greater than or equal to the observed value, but exceeds the APE otherwise.

It's worth pointing out that due to the introduction of regularization, WIS differs from the penalized quantile regression loss used to train our estimation models. For model evaluation, we aggregate WIS scores by averaging over all reference dates $t$ and locations $i$ while considering log-scale quantities. This approach leverages the geometric mean, which provides a more accurate assessment of positively skewed relative errors.

## Adaptive modeling protocol

The correction of real-time data revisions involves repeatedly forecasting target values using epidemic quantities observed up to a given estimation date, denoted by $s_0$. At each estimation date, we simulate the real-time setting by training the model using the latest available revisions of past values. Specifically, the model is provided with the following set

of inputs for a given location $i$:

$$
\begin{array}{cccccccc}
& Y_{i,s_0,0} & Y_{i,(s_0-1),1} & Y_{i,(s_0-2),2} & \cdots & Y_{i,(s_0-L+1),(L-1)} & Y_{i,(s_0-L),L} & \cdots \\
\text{Reference date:} & s_0 & s_0-1 & s_0-2 & \cdots & s_0-L+1 & s_0-L & \cdots \\
\text{Revision index:} & 0^{\text{th}} & 1^{\text{st}} & 2^{\text{nd}} & \cdots & (L-1)^{\text{th}} & L^{\text{th}} & \cdots
\end{array}
$$

These values represent the most recent revisions of past observations that would have been available at $s_0$. For example, $Y_{i,s_0,0}$ is the initial report for reference date $s_0$, $Y_{i,(s_0-1),1}$ is the second revision for reference date $s_0-1$, and so on. As the estimation date progresses from $s_0 - 1$ to $s_0$, the data revision sequence

$$
Y_{i,(s_0-L),0:L} = \{Y_{i,(s_0-L),0}, Y_{i,(s_0-L),1}, Y_{i,(s_0-L),2}, \dots, Y_{i,(s_0-L),L}\}
$$

is newly added to the training set, while the forecast made for the reference date $s_0-L$, based on the 0th through $(L-1)$th revisions, can now be evaluated since the target has become available. Data for reference dates $t$ such that $s_0-L < t \leq s_0$ continue to serve solely as input covariates to generate real-time forecasts, until their corresponding targets become available.

To select hyperparameters $(c, \lambda, \gamma)$, we perform a grid search with 3-fold cross-validation [29]. At each combination of hyperparameter values, the training set is partitioned into three subsets; in each fold, the model is trained on two subsets and validated on the third. The process is repeated so that each subset serves once as the validation set. Validation performance is evaluated using the average WIS across all reference dates, and the hyperparameter configuration that minimizes this score is selected.

## Experimental results

In this section, we apply the proposed framework to multiple state-level, daily COVID-19 datasets obtained from the Delphi Epidata API [22]. These datasets differ in source, structure, and update frequency, and each displays distinct revision patterns, posing challenges for direct performance comparisons across models. Each dataset also exhibits a distinct data revision pattern, making direct comparisons of model performance across datasets nontrivial.

Beyond evaluating the performance of our proposed method, we compare it against established approaches such as Epinowcast and NobBS in terms of both forecast accuracy and computational efficiency for count-type public health data. We further demonstrate the adaptability of our framework by extending it to weekly datasets, including dengue and ILI case counts.

All data (including properly versioned datasets) and code used in our analysis are publicly available at: https://github.com/cmu-delphi/data-revisions-forecasting-paper.

### Description of datasets

**Insurance claims:** We use insurance claims data provided by Change Healthcare (CHNG). CHNG aggregates claim data from numerous healthcare providers and payers, and the information provided by CHNG spans more than two thousand of the most populous US counties, covering more than 45% of the total US population. Our analysis focuses on a time series comprising the aggregate count of claims featuring ICD codes indicative of COVID-19 diagnoses recorded daily within each county. The reference date corresponds to the claim's date of service, while the report date denotes its appearance in the CHNG database, which may vary considerably depending on providers' claim filing times. Sometimes, Delphi receives the initial report release for a reference date on the same day. We produce forecasts for each report date between 2021-06-01 and 2023-01-10 (a total of 589 days including the Delta wave and the Omicron wave of COVID-19).

**Antigen tests:** This dataset is provided by Quidel Corporation (Quidel), which supplies devices used by healthcare providers to conduct COVID-19 antigen tests. We construct a time series of the fraction of positive tests using this

dataset. The test records indicate the test date (when the test was conducted) and storage date (when the test was logged into Quidel's MyVirena cloud storage system). The test date serves as the reference date, and test records with a storage date preceding the test date or more than 90 days after are excluded. The report date is defined as the date the records are shared with the Delphi Group via a cloud platform. We produce forecasts for all the states and report dates ranging from 2021-05-18 to 2022-12-12 (a total of 574 days).

**COVID-19 cases in MA:** The Massachusetts Department of Public Health (MA-DPH) provides a comprehensive revision history of COVID-19 case reporting [30]. This dataset includes the 7-day moving average of confirmed COVID-19 cases, updated daily from 2021-01-01, until 2022-07-08. After this date, the reporting frequency transitioned to weekly updates, occurring every Thursday. The initial release of the report for a reference date $t$ is usually made on date $t +$ 1. Unlike the other two datasets—whose fractional denominators exhibit noticeable revision sequences similar to their numerators—a distinctive feature of this dataset is that the COVID-19 confirmed cases are typically normalized by population figures that are sufficiently large to render temporal fluctuations negligible relative to the numerators. We used data reported before 2022-07-08, and produced forecasts for report dates ranging from 2021-07-01, to 2022-06-24 (a span of 359 days).

## Data processing

**Data filtering:** All datasets are filtered based on the specified time period to ensure data quality. For CHNG outpatient insurance claim data and Quidel antigen test data, this filtering process excludes periods affected by prolonged data reporting issues, such as significant declines in report volume over several months. These anomalies, often manifesting as abrupt shifts in the data distribution, are more indicative of data quality issues than of genuine changes in revision patterns. The MA-DPH case data are filtered to include only the period with consistent daily reporting. As shown in Fig 2, over 90% of the confirmed cases for a given reference date are reported within 7 days. Given this pattern, developing a data revision forecasting model for weekly reports is unnecessary.

## Experimental setup

**Selection of target lag $L$:** To better understand the data revision patterns, we analyze the distribution of the variable $p_{itl}$, defined as $p_{itl} = Y_{itl}/Y_{itL_{it}}$, across reporting lags. For count-type data (e.g., the number of confirmed cases), $p_{itl} \times 100\%$

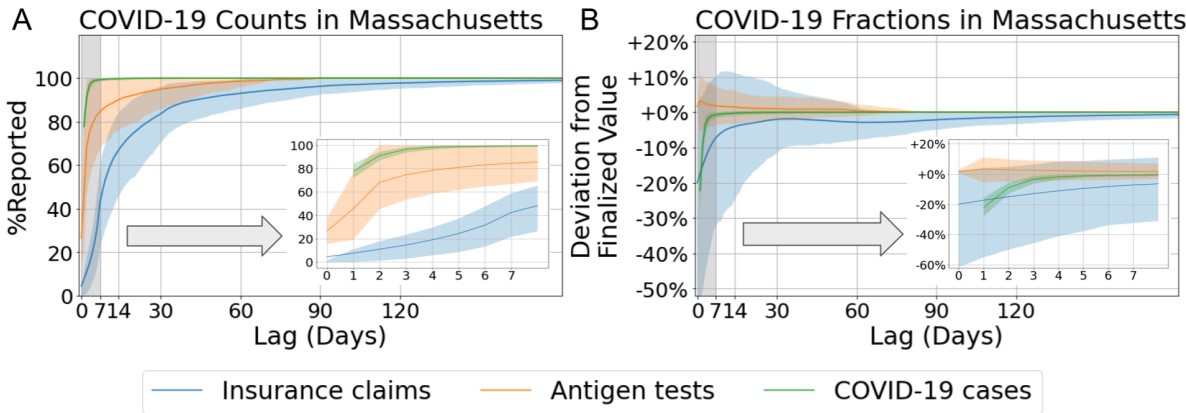

**Fig 2**. **Data revision patterns for different indicators.** (A) Mean percentage of counts reported relative to the values revised 300 days later, averaged over all reference dates and plotted by reporting lag for Massachusetts. Shaded bands represent the 10th to 90th percentile interval. (B) Mean values of COVID-19-related fractions normalized by their corresponding revised values after 300 days, also averaged over all reference dates and plotted by lag for Massachusetts.

quantifies the percentage of the total value reported at lag $l$ for location $i$ and reference date $t$. For fraction-type data (e.g., the fraction of COVID-19 insurance claims), $p_{itl}$ represents the normalized provisional estimate. Since the finalized value $Y_{itL_{it}}$ is not observable in real time, we temporarily approximate it using a sufficiently large lag of 300, under the assumption that the revision process has effectively converged by that point. Specifically, we set $L_{it} = 300$, $Y_{itL_{it}} = Y_{it,300}$ to serve as a practical surrogate for the finalized target value.

The distribution of $p_{itl}$ provides insight into how data revision sequences evolve over time. In addition to the apparent day-of-week effect and week-of-month effect, the efficiency of data revision is significantly influenced during periods when the epidemic curve is at or near its peak. Fig 3 illustrates this phenomenon using CHNG outpatient insurance claim data in MA as an example. Overall, the revision of COVID-19 claims exhibit greater variance than non-COVID claims, underscoring the difficulty of the forecast task.

Although there may be a considerable degree of heterogeneity in $L_{it}$ (the target horizon for $Y_{it}$, example shown in S1 Fig), the most substantial revisions are typically made within the first two months for the majority locations including states and populous counties for CHNG outpatient insurance claims data. The bottom panel of Fig 3 shows an example based on CHNG outpatient COVID-19 insurance claims data. It reveals that, for states in HHS Region 1 and Region 2, almost all mean %reported values for COVID-19 reach 90% when the lag equals 60 days.

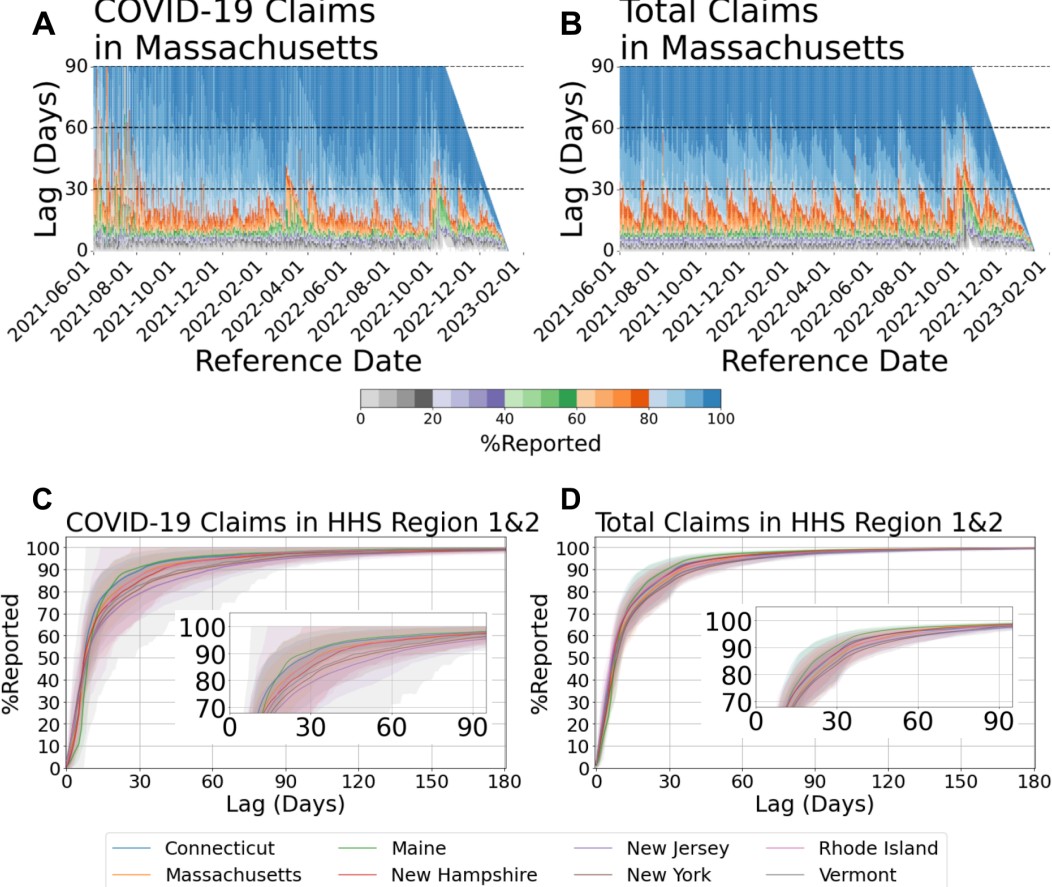

**Fig 3. Revision patterns of COVID-19 claims and total claims** (A) COVID-19 claims in Massachusetts and (B) total claims in Massachusetts, shown as heatmaps of the proportion reported by reference date and reporting lag. (C) COVID-19 claims and (D) total claims across HHS Regions 1 & 2, showing cumulative reporting curves by state. Shaded bands represent the 10th to 90th percentile interval.

In our experiments, we set $L = 60$ for the CHNG outpatient COVID-19 insurance claims data and $L = 45$ for the Quidel COVID-19 antigen tests data, both applied uniformly across all states considered. For the data on confirmed cases from MA-DPH, we set $L = 14$, ensuring that at least 90% cases are reported while keeping $L$ relatively small.

**Training frequency:** We generate state-level forecasts following the adaptive modeling protocol described in section 3.2. To improve computational efficiency, model training is performed every 30 days, except for the MA-DPH confirmed case forecasts, where the shorter target lag of 14 days requires retraining every 7 days. While this adjustment reduces computational cost, it may degrade predictive performance in the presence of non-stationarity—particularly in scenarios involving abrupt and substantial changes in data revision patterns—as less frequent retraining limits the model's ability to adapt in a timely manner. On each retraining date $s_{\text{train}}$, the model is updated with newly available training data and subsequently used to generate quantile forecasts for all epidemic quantities reported on dates $s \in [s_{\text{train}}, s_{\text{train}} + 30)$ (or $s \in [s_{\text{train}}, s_{\text{train}} + 7)$ for the MA-DPH case forecasts).

**Location-specific model training:** Both the CHNG outpatient insurance claims data and Quidel antigen tests data are subject to geographic variation in market share, health-seeking behavior, and reporting practices. These differences render the data incomparable across locations and result in location-specific revision patterns. In this study, we do not attempt to address spatial heterogeneity; instead, we fit the model separately for each location.

## Accuracy of revision forecasts

In the subsequent analysis, we evaluate forecasting performance using the Weighted Interval Score (WIS). To reiterate, the WIS measures the distance between the forecast distribution and the observed target value on the log scale. The quantity $\exp(\text{WIS}) - 1$ provides an interpretable approximation of the absolute relative error. The arithmetic mean of WIS captures relative error in log space, which is equivalent to the geometric mean in the original scale.

The forecasting performance is evaluated relative to a baseline model, defined as a flat-line predictor whose forecasted median is the 7-day moving average of the most recent observations. Consequently, the WIS for the baseline reduces to the absolute error on the log scale between the most recent observation and the finalized.

We evaluate the forecasting performance of our framework across the three datasets described in section 4.1. For the MA-DPH confirmed cases, which are usually normalized by a constant (the MA population), we generate forecasts only for the counts. For the insurance claims, we produce forecasts for both the counts and the fractions of COVID-19–related outpatient claims. For the antigen tests, we forecast the fraction of positive tests among all tests conducted.

The following is a summary of the experimental results:

- Our data revision forecasting framework substantially reduces forecast error, particularly at shorter lags (e.g., within the first 0–5 days). However, the marginal improvement diminishes as the lag increases. These results suggest that modeling and forecasting data revisions is most beneficial in settings where timely estimates are needed in near real-time.
- Comparing across the three datasets, we find that the task is most difficult for the insurance claims data, followed by antigent tests, and finally MA-DPH confirmed cases. Intuitively, this ordering matches Fig 2, where the claims data exhibit the slowest convergence among the three.
- Abrupt distributional shifts remain a significant challenge. Our model relies on historical data revision patterns to forecast future updates, which implicitly assumes that these patterns are stationary over time. When the revision process undergoes a sudden and substantial change—particularly one that has not been observed in the training data—the model may struggle to adapt, resulting in degraded forecasting performance.

Next, we demonstrate the forecasting performance of our model and how the performance varies along difference dimensions.

## Aggregate accuracy by lag

We investigate how forecasting performance varies as a function of the lag at which the report (or the revision) is made. For a given report date $s$, we define the forecasting task as having a lag of $s-t$ when predicting $Y_{itL}$ using all data available up to and including date $s$. Since the revision sequence of $Y_{it}$ gradually converges to its finalized value, forecasts made with shorter lags are inherently more challenging due to the limited availability of information in earlier stages.

Figs 4 and 5 present the evaluation results for count and fraction forecasts, respectively, stratified by lag and averaged over all locations and reference dates. For confirmed cases from MA-DPH (Fig 4A), the mean WIS of the baseline model begins at approximately 2.56 when the lag is 1, corresponding to a mean absolute relative error of roughly 1189.43% under this evaluation metric. In contrast, the mean WIS of our distributional forecasts is substantially lower—0.12 (approximately 13.04% absolute relative error) when using a 180-day training window, and 0.14 (approximately 15.55%) when using a 365-day training window. These results demonstrate that our approach outperforms the baseline, particularly when the lag is less than 4 days.

The performance gap is even more pronounced for the insurance claims data, where revision patterns tend to be more frequent and variable. As shown in the right panel of Fig 4, the baseline model maintains a mean WIS above 0.18—corresponding to an absolute relative error of approximately 20%—even after 14 days of revision. In comparison, our model yields a mean WIS of 0.23 (approximately 25.65% absolute relative error) at lag 0 when using a 180-day training window, and 0.20 (approximately 22.65%) with a 365-day training window. After 14 days of revision, the forecasting accuracy improves with our model achieving a mean WIS of 0.12 (approximately 12.70% absolute relative error) using the 180-day window, and 0.10 (approximately 10.80% absolute relative error) using the 365-day window.

Similarly, Fig 5 illustrates the evaluation results of COVID-19 fraction forecasts as a function of lag. For insurance claims data, the mean WIS exceeds 0.45 which approximates an absolute relative error of around 56.83% when comparing the initial release (lag = 0) to the target (lag = 60). However, this mean WIS is reduced to around 0.16 (approximately 16.79% absolute relative error) using our distributional forecasts with a 180-day training window and a mena WIS

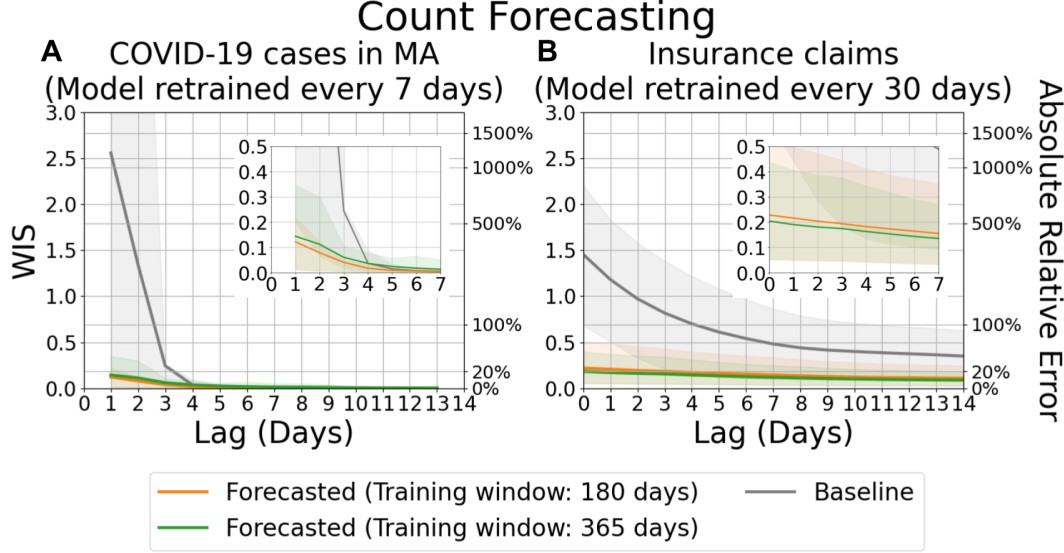

**Fig 4. Evaluation of forecasts for counts, aggregated by lag.** (A) Forecasts of finalized confirmed COVID-19 case counts in MA. (B) Forecasts of COVID-19 insurance claims across all states, based on CHNG outpatient insurance claims data. Solid lines indicate the mean WIS, which approximates absolute relative errors between the most recent report and the target, averaged over locations and reference dates for each lag. Shaded areas represent the 10th to 90th percentile interval.

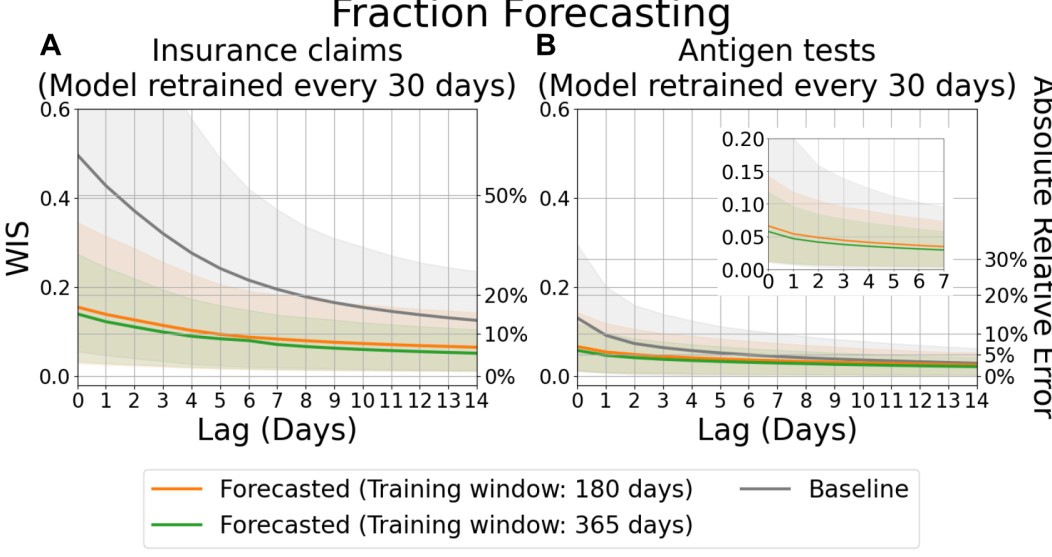

**Fig 5. Evaluation of forecasts for fractions, aggregated by lag.** (A) Forecasts of the fraction of COVID-19 insurance claims based on CHNG outpatient insurance claims data. (B) Forecasts of the fraction of positive COVID-19 antigen tests based on Quidel antigen tests data. Solid lines represent the mean WIS, , which approximates absolute relative errors between the most recent report and the target, averaged over locations and reference dates for each lag. Shaded areas indicate the 10th to 90th percentile interval.

of 0.14 (approximately 15.00% absolute relative error) with a 365-day training window. Even after 7 days of revisions, the distributional forsecasts continue to yield substantial improvements.

In contrast, the antigen tests are considerably less affected by the data revision problem. Nevertheless, even when provisional reports closely approximate the target, our framework still achieves substantial improvements in forecast accuracy. Specifically, at the initial release (lag = 0), the mean WIS decreases from approximately 0.13 (corresponding to a 13.95% absolute relative error) to 0.07 (6.88% absolute relative error) when using a 180-day training window, and further to 0.06 (5.97% absolute relative error) when using a 365-day training window.

## Aggregate accuracy by reference date

The difficulty of the forecasting task varies not only with lag but also over time. Figs 6 and 7 present evaluation results for forecasts made for reference dates from 2021-06-01 to 2022-03-01 at a fixed lag of 7 days, corresponding to count and fraction targets, respectively. The results are stratified by reference date and averaged across all locations. In each figure, the color strip (i.e., a one-dimensional heatmap below the line plot) indicates the target values over time.

Across most of the study period, the model produces stable and accurate forecasts, reflected in consistently low WIS values (typically below 0.14, corresponding to less than 15% absolute relative error) outside of periods of extreme epidemiological change. Forecast accuracy does decline at times, most notably during the Omicron wave (November 2021 to February 2022). This degradation arises from the lagged nature of the model: coefficients estimated from data observed $L$ days earlier cannot fully capture abrupt shifts in revision patterns, leading to reduced performance under non-stationary conditions.

The Omicron wave, the largest observed during the COVID-19 pandemic, was marked by unprecedented infection rates and severe strain on public health reporting systems. Unlike the preceding Delta wave (July to November 2021), the Omicron surge introduced abrupt and substantial changes in revision dynamics. These shifts posed a significant challenge for the model, which struggled to adapt to revision patterns not previously encountered in training data.

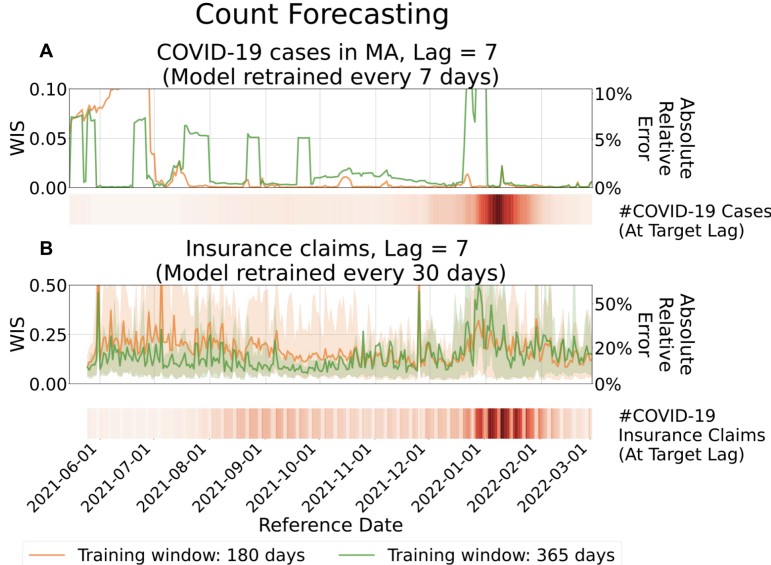

**Fig 6. Evaluation of forecasts for counts, aggregated by reference date** Top: Forecasts of finalized confirmed COVID-19 case counts in MA. Bottom: Forecasts of COVID-19 insurance claims across all states, based on CHNG outpatient insurance claims data. Solid lines represent the mean WIS at lag 7, averaged over locations for each reference date. Shaded areas indicate the 10th to 90th percentile interval. The accompanying heatmaps display the corresponding target values, with darker shades indicating larger number of cases or claim counts.

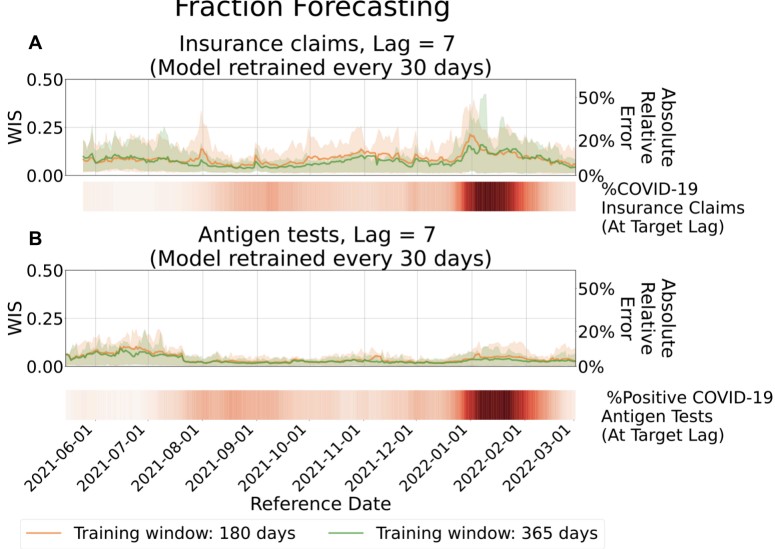

**Fig 7. Evaluation of forecasts for fractions, aggregated by reference date.** Top: Forecasts of the fraction of COVID-19 insurance claims based on CHNG outpatient insurance claims data. Bottom: Forecasts of the fraction of positive COVID-19 antigen tests based on Quidel antigen tests data. Solid lines represent the mean WIS at lag 7, averaged over locations for each reference date. Shaded areas indicate the 10th to 90th percentile interval. The accompanying heatmaps display the target values, with darker shades indicating higher fractions.

Another challenging period occurred from June to July 2021, when infection rates were extremely low. Because WIS is based on absolute deviations between forecasts and targets, this metric tends to exaggerate relative errors when the target values are very small.

A comprehensive set of state-level evaluation results for count revision forecasts based on insurance claims data is presented in S1 Appendix. The findings are consistent with the main analysis: model performance remained stable throughout most of the study period but degraded during periods of major epidemiological change, such as the Omicron wave. Some heterogeneity in absolute WIS values was observed across states, reflecting variation in population size and reporting practices.

### Impact factors of forecast accuracy

Our method exhibits degraded performance under two specific scenarios: (1) periods marked by abrupt changes in the target surveillance trend, and (2) periods when the target values are extremely small. The first scenario relates to the direction of the trend in the target surveillance curve, while the second pertains to the magnitude of the target values. We now examine the distribution of WIS across different lags, conditioned separately on each of these two factors, using the CHNG outpatient COVID-19 fraction data as a case study.

Fig 8A shows the distribution of WIS across different lags, stratified by the trend direction of the CHNG outpatient COVID-19 fraction. We classify each instance into one of three categories: increasing ("up"), decreasing ("down"), or stable ("flat") trends. The trend indicator $Z_{it}$ is defined as:

$$Z_{it} = \begin{cases} 1, & \text{if } \frac{\widetilde{Y}_{itL}}{\widetilde{Y}_{i(t-7)L}} \geq 1.25 \\ -1, & \text{if } \frac{\widetilde{Y}_{itL}}{\widetilde{Y}_{i(t-7)L}} \leq 0.75 \\ 0, & \text{otherwise} \end{cases}$$

We assign $Z_{it} = 1$ if the 7-day average of the target value has increased by at least 25% relative to the previous week, indicating an upward trend for location $i$ at date $t$. Conversely, $Z_{it} = -1$ denotes a decrease of at least 25%, indicating a downward trend. All other cases are classified as flat ($Z_{it} = 0$).

Forecasting performance notably improves during periods with minimal changes in the target surveillance curve. The performance is the poorest during "Down" periods for quantities with only 0–7 revisions. This can be attributed to the fact that the "Down" category primarily corresponds to the downswing of the Omicron wave, whereas the "Up" category includes reference dates from the upswings of both the Delta and Omicron waves (as shown in S2 Fig). Overall, the model performs better during the Delta wave than during the Omicron wave, as the magnitude of distributional shift in the

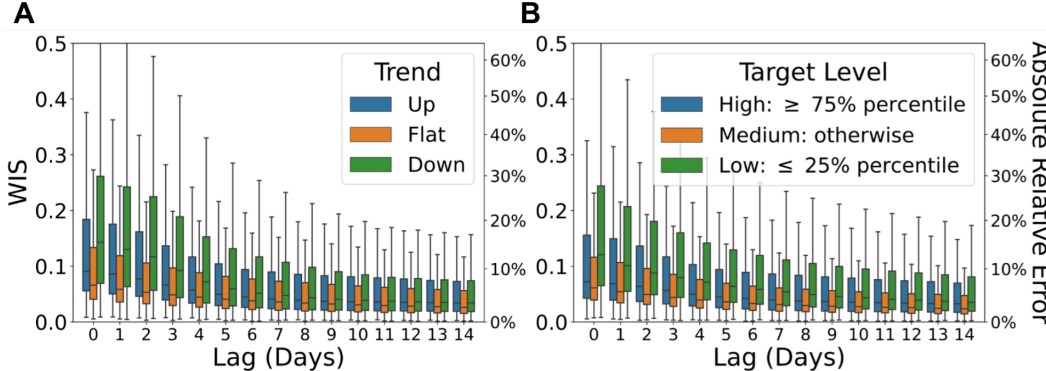

**Fig 8**. **Boxplots illustrating the impact of surveillance conditions on forecast accuracy** (Each box displays the 25th, 50th (median), and 75th percentiles of the WIS). (A) Forecasts stratified by the direction of the target surveillance trend—"Up", "Flat", or "Down". (B) Forecasts stratified by the magnitude of the target, categorized as "High", "Medium", or "Low".

data revision pattern during Delta was comparatively smaller. After the first 7 revisions, the performance gap across the three trend categories narrows, with the performance ranking shifting to: "Flat", "Down" and then "Up".

Fig 8B illustrates the distribution of WIS over lags, stratified by whether the target surveillance value falls into the categories of "High", "Medium", or "Low". A target value is classified as "High" if it is greater than or equal to the 75% percentile, while it is classified as "Low" if it is less than or equal to the 25% percentile. The performance order, from best to worst, consistently ranks as "Medium", "High", "Low" across lags. Notably, even after the first 14 revisions, the performance gap across these three categories remains significant.

## Comparison of performance with alternative methods

In this section, we demonstrate that our model achieves forecast accuracy comparable to, or exceeding, that of NobBS [1] and Epinowcast [14], while substantially reducing computational runtime. These two methods were selected for comparison as they are among the most established in the literature, widely used in public health research projects, and supported by well-maintained R packages.

Since both methods are specifically designed for count-type data, our comparison is limited to count-type datasets. For the COVID-19 insurance claims with daily observations, we use a 180-day training window and a target lag of 60 days. To ensure a fair comparison, we apply the same 180-day moving window and a maximum delay of 60 days to NobBS and Epinowcast. To manage computational demands while maintaining consistency across models, we train all methods—including Delphi-RF—at 30-day intervals. Additionally, we evaluate performance on daily confirmed cases from MA-DPH, where revisions stabilize more quickly. For this dataset, we adjust the training frequency to every 7 days, set the maximum delay to 14 days, and maintain the 180-day moving window. In Delphi-RF, the target lag is also set to 14 days to align with these adjustments.

We further extend our comparison to weekly data. To ensure compatibility with weekly data, covariates containing daily change information were excluded. The full model is expressed as:

$$
\begin{aligned}
Q^{\tau}_{f(Y_{itL}))|X_{itl}} \\
= X_{itl}\beta^{\tau} \\
= \beta_0^{\tau} + f(Y_{itl})\beta_1^{\tau} \quad &\text{(Intercept, disease activity level)} \\
+ \mathbf{I}_{\text{first-week}(t+l)}\beta_2^{\tau} \quad &\text{(week-of-month effects)} \\
+ \left(f(Y_{i(t-7)(l+7)}) - f(Y_{i(t-7)l_{\min}})\right)\beta_3^{\tau} \quad &\text{(Recent revision magnitude, } t-7) \\
+ \left(f(Y_{i(t-14)(l+14)}) - f(Y_{i(t-14)l_{\min}})\right)\beta_4^{\tau} \quad &\text{(Recent revision magnitude, } t-14) \\
+ f(Y_{i(t-7)(l+7)})\beta_5^{\tau} + f(Y_{i(t-14)(l+14)})\beta_6^{\tau} \quad &\text{(Short-term epidemic trends)}
\end{aligned}
$$

where $Y_{itl}$ represents the counts reported for the week spanning reference dates $t$–6 to $t$, as of the report date $t + l$, for location $i$.

To further evaluate model performance across diverse surveillance settings, we test the models on two additional datasets with distinct characteristics to assess their robustness. First, we apply the models to Puerto Rico (PR) dengue weekly surveillance data spanning from 1991-12-23 to 2010-11-29 (989 weeks). This dataset features a long historical record and strong seasonality, differs from the more irregular trends observed in COVID-19 data. Applying our method to the dengue data enables assessment of its ability to capture seasonal dynamics and long-term surveillance patterns. The target lag is set to 10 weeks, with 104 weeks of data used for training. For comparison, weekly forecasts are generated using NobBS and Epinowcast over the same time period. The maximum reporting delay is set to 10 weeks, and a

104-week moving window is applied, consistent with the setup in [1]. For all three models, training and forecasting were conducted on a weekly basis.

We also test the models on national weekly influenza-like illness (ILI) case counts from 2014-06-30 to 2017-09-25 (170 weeks), which follow a distinct reporting pattern. For Delphi-RF, a 27-week training window is used with a target lag of 26 weeks. NobBS and Epinowcast are similarly configured with a 26-week maximum delay and a 27-week moving window, following the same settings in [1].

All experiments were conducted on an Apple Mac Mini equipped with a 3.0 GHz 6-core Intel Core i5 processor, running R version 4.4.2.

As shown in Fig 9, our model delivers accurate forecasts across all evaluated datasets. For daily COVID-19 signals, Delphi-RF consistently outperforms NobBS and achieves accuracy comparable to that of Epinowcast. To ensure a fair comparison across methods, a fixed 180-day training window is used—a conservative choice made to accommodate the computational demands of more resource-intensive methods such as Epinowcast. This restriction, however, can be sub-optimal for Delphi-RF. In fact, forecast accuracy improves for certain reference dates and locations when the training window is extended (e.g., FL and NJ; see Fig I and Fig AE in S1 Appendix), with only a modest increase in computation time.

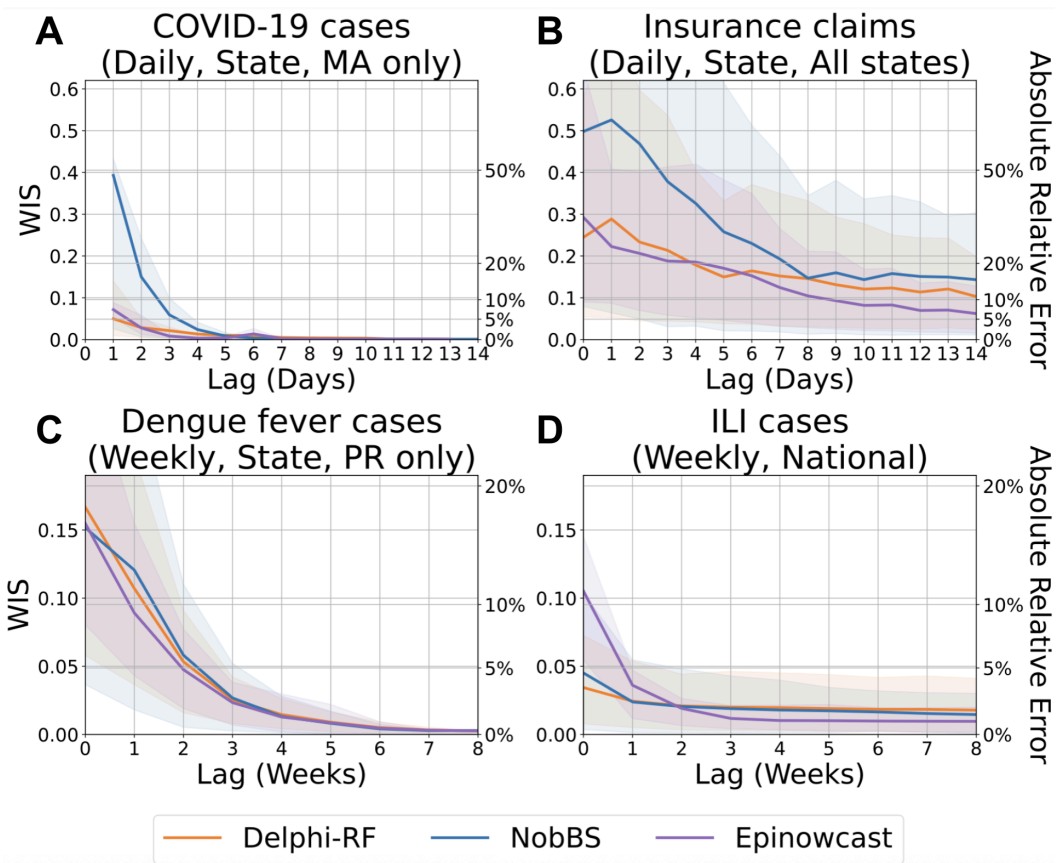

**Fig 9. Comparison of count forecast evaluation results with NobBS and Epinowcast.** (A) Forecasts of finalized confirmed COVID-19 case counts in Massachusetts. (B) Forecasts of finalized COVID-19 insurance claim counts across all states based on CHNG outpatient data. (C) Forecasts of dengue fever case counts in Puerto Rico. (D) Forecasts of ILI case counts nationwide. Solid lines represent the mean WIS, which approximates absolute relative errors between the most recent report and the target, averaged over locations and reference dates for each lag. Shaded areas indicate the 10th to 90th percentile interval.

For example, increasing the window to 365 days results in less than a twofold increase in runtime for both the Insurance Claims and MA-DPH COVID-19 case data (S1 Table). In contrast, Epinowcast and NobBS become computationally infeasible under the same setting, with Epinowcast exceeding the 30-minute runtime cutoff for a single location–report-date pair. When applied to Insurance Claims data across all 50 states, sequential training using either method with a 180-day or longer window would require more than two days—rendering them impractical for daily forecasting tasks.

For weekly data, Delphi-RF exhibits competitive forecasting performance for dengue fever cases in Puerto Rico. In the case of national ILI counts, Delphi-RF outperforms both benchmark methods at lag 0. Although Delphi-RF does not outperform Epinowcast when the reporting lag exceeds 3 weeks, the absolute relative errors for all methods are sufficiently low—consistently below 2.5%—rendering performance differences practically negligible.

The runtime reported in Table 1 reflects both the training and testing phases required by our model (including the computing time for data pre-processing), which is substantially faster than the other two methods. Unlike Epinowcast and NobBS, which require simultaneous training and forecasting, our model benefits from the modularity of machine learning frameworks, allowing for independent training and inference. Once trained, the model can be repeatedly applied to generate forecasts for new data. This flexibility allows users to tailor the training frequency to operational constraints, a feature not available in Epinowcast or NobBS. The efficiency also enables our model to produce revision forecasts for multiple signals at different temporal resolutions in real time while requiring significantly fewer computational resources.

## Ablation study

To assess the importance of our modeling choices described in the Methods section, we conducted an ablation study in which individual feature groups were removed from the model and the resulting performance was evaluated in terms of WIS. Fig 10 compares the performance of the full-feature model to the ablated variants. WIS is averaged across all locations and reference dates, and results are stratified by lag, with standard errors of the mean shown as error bars.

Across all datasets, the full-feature model generally outperforms the ablated models, confirming that each feature group contributes complementary signal. The figure also highlights that feature contributions are both lag-dependent and data-dependent. For example, in the dengue fever and ILI weekly datasets, omitting the 7-day moving average produces the largest degradation in performance. This is because, unlike daily data where revision magnitude features provide complementary information about disease evolution, the weekly setting relies heavily on the 7-day moving average to represent disease activity. In contrast, for COVID-19 case counts in Massachusetts (Fig 10C), we observe that at a few specific lags, some ablated models perform comparably or even slightly better than the full-feature model. This is due to a combination of sampling variability and feature redundancy: certain predictors (e.g., lag values and smoothed averages)

**Table 1**. **Computing time comparison across methods and datasets.** Computing time required by different methods applied to various datasets, measured per location and per report date. The table presents the mean and standard error of the mean (SEM) for computing time. For daily data, all models are trained and generate forecasts every 30 days for CHNG outpatient insurance claims and every 7 days for MA-DPH COVID-19 confirmed cases. For weekly data, models are trained and generate forecasts on a weekly basis. To ensure a fair comparison, all settings—including maximum delay and training window size—are kept the same across methods.

| Computing Time(s) (per location per report date) | | Model | | | |
|---|---|---|---|---|---|
| | | Delphi-RF Training (once/week or month) | Delphi-RF Testing | Epinowcast | NobBS |
| Dataset | Confirmed Cases (Daily, State, MA only) | $6.773 \pm 0.018$ | $0.369 \pm 0.006$ | $406.097 \pm 16.190$ | $24.220 \pm 0.675$ |
| | Insurance Claims (Daily, State, All states) | $23.712 \pm 0.029$ | $0.819 \pm 0.008$ | $2386.512 \pm 230.895$ | $96.012 \pm 0.453$ |
| | Dengue Fever Cases (Weekly, State, PR only) | $6.848 \pm 0.106$ | $0.153 \pm 0.008$ | $64.628 \pm 0.395$ | $8.337 \pm 0.033$ |
| | ILI Cases (Weekly, National) | $2.006 \pm 0.032$ | $0.136 \pm 0.003$ | $18.373 \pm 2.139$ | $5.960 \pm 0.055$ |

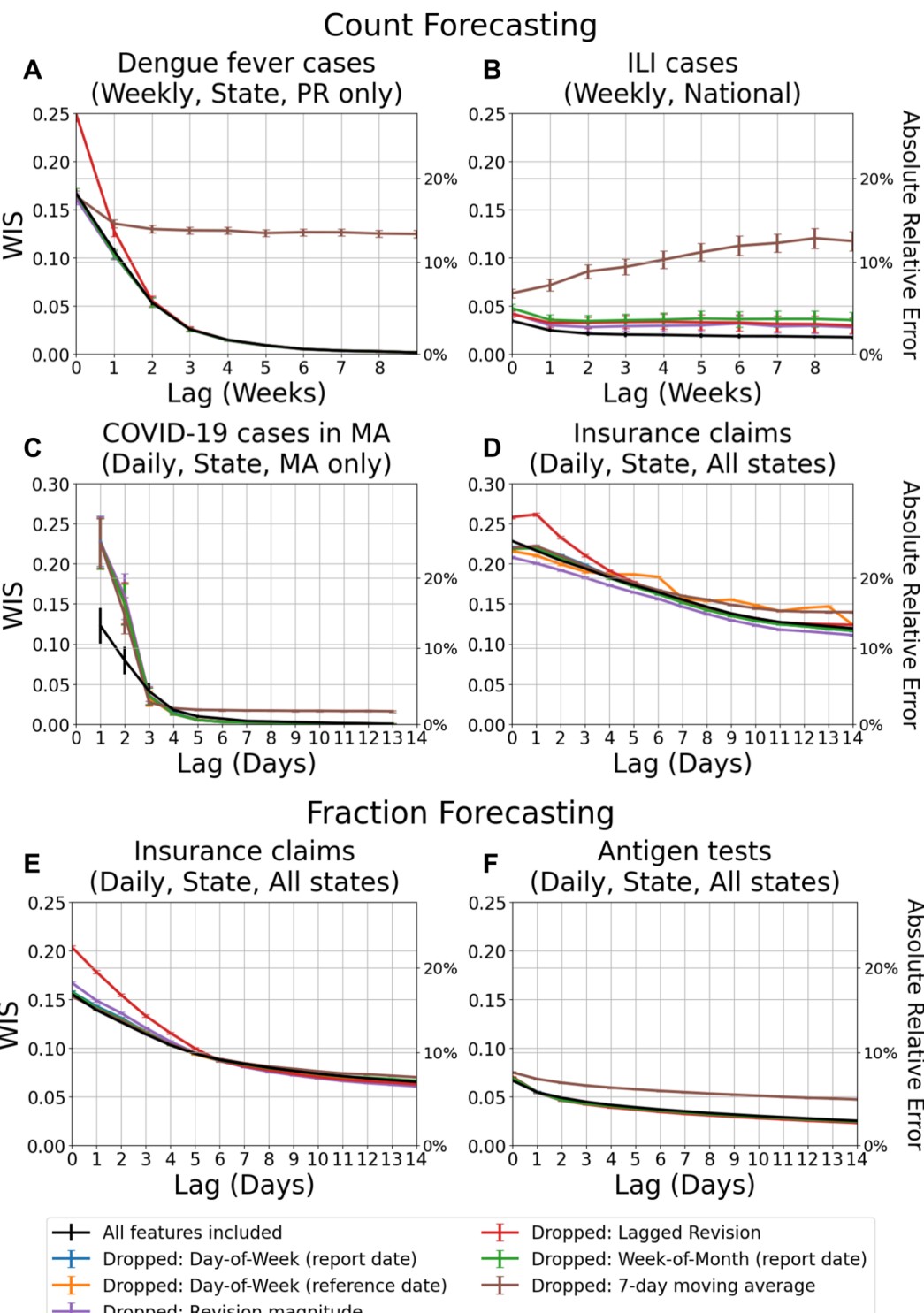

**Fig 10. Ablation study of Delphi-RF features for forecasting performance.** Each colored curve represents the performance when a specific feature group is dropped from the model (e.g., day-of-week effect, week-of-month effect, lagged values, revision magnitude, or 7-day average). The black curve shows the baseline model that includes all features. Error bars represent the standard error of the mean WIS. The right-hand y-axis shows the corresponding absolute relative error percentage. Lower values indicate better predictive performance.

encode overlapping information, so removing one can occasionally reduce noise without materially degrading predictive signal. Moreover, the relative importance of features varies across horizons, so at isolated lags a reduced feature set can temporarily appear more favorable.

Another practical point concerns the use of lag padding. For insurance claims (both counts and fractions) and antigen tests, we employ a lag pad of 1. This strategy provides richer training information by exposing revision patterns across lags, but it also requires careful alignment of lag indices. At very short lags, the influence of inverse lag features dominates, though this effect diminishes quickly as lag increases. For antigen test fraction forecasting, the 7-day moving average also plays an important stabilizing role. By comparison, the other features—day-of-week, week-of-month, and revision magnitude—still contribute, but are relatively less informative in this setting.

### Hyperparameter sensitivity analysis

There are four hyperparameters in the Delphi-RF framework—regularization strength ($\lambda$), training-window length ($W$), lag padding ($c$), and decay parameter ($\gamma$)—as described in the Methods. To assess their influence, we perform a sensitivity analysis.

We adopt $\lambda = 0.1$, $\gamma = 0.1$, and $W = 180$ days as the base configuration. For lag padding, the base is $c = 0$ for dengue fever cases, ILI cases, and COVID-19 case counts in MA, and $c = 1$ for insurance claims and antigen tests. The latter two exhibit comparatively slower revision convergence, so including a one-step lag neighborhood ($c = 1$) allows the model to exploit informative early-lag patterns that can improve short-horizon forecasts. From this base, we vary one hyperparameter at a time while holding the others fixed.

Sections 4.5 and 4.6 compare training-window choices. In aggregate, $W = 180$ $and$ $W = 365$ yield similar accuracy; however, as shown in section 4.8, longer windows can improve performance for specific locations and reference periods when revision dynamics are stable, whereas under distribution shift they may introduce outdated information and reduce accuracy.

S3 Fig and S4 Fig present performance across $\lambda$ and $\gamma$ at lags $l_{min}$, 7, and 14 days ($l_{min} = 1$ for COVID–19 confirmed cases in Massachusetts and $l_{min} = 0$ for the other datasets). In general, the results indicate that forecasts are not highly sensitive to the choice of these two hyperparameters, with $\lambda = 0.1$ and $\gamma = 0.1$ performing well across most settings. An exception is observed for COVID-19 confirmed cases in MA, which exhibit broad sensitivity to both $\lambda$ and $\gamma$. Because revisions are concentrated within the first week, early-lag patterns (0–2 days) are sharp and high in magnitude, making forecasts particularly dependent on whether the model can capture this fine structure without overfitting. Dengue fever cases display a different profile. The initial report (lag 0) is relatively stable across seasons and strongly shaped by multi-year seasonal signals, rendering forecasts at this horizon comparatively robust to the choice of $\lambda$ and $\gamma$. At subsequent lags (1–2 weeks), however, revisions remain sizable and heterogeneous, and sensitivity emerges as regularization and decay directly determine how much weight the model assigns to these early but volatile revisions.

S5 Fig summarizes performance across $c$. For datasets with slower revision processes, such as antigen tests and insurance claims, padding helps at short lags (e.g., 0–1 day), where borrowing strength across adjacent lags improves forecast stability. However, the benefit diminishes—and can even reverse—at longer horizons (e.g., 7–14 days), where pooled lags differ more strongly. By contrast, for datasets with faster revision completion, such as Massachusetts confirmed cases and dengue fever cases in Puerto Rico [1], early-lag revision patterns are sharply distinct. In these settings, padding ($c > 0$) introduces noise and degrades accuracy. Sensitivity decreases once revisions have stabilized at larger lags.

In general, hyperparameter effects in Delphi-RF are signal- and lag-dependent, underscoring the need to align hyperparameter choices with the revision dynamics of each dataset.

## Conclusion and discussion

This paper introduces a comprehensive modeling framework, Delphi-RF, designed to capture data revision dynamics and generate distributional revision forecasts in real-time. The application of our model extends to diverse public health data sources, encompassing outpatient COVID-19 claims data, COVID-19 antigen test data, and confirmed cases from MA-DPH.

Delphi-RF produces accurate and adaptive forecasts for target surveillance values. These forecasts are particularly valuable for auxiliary epidemiological data streams, which are frequently used as predictive features in real-time epidemic forecasting but are often affected by the problem of data revisions. Delphi-RF competes with more elaborate nowcasting frameworks (e.g., EpiNowcast) because it uses quantile regression forests to learn nonlinear effects from a set of covariates that encode reporting behavior (day-of-week and week-of-month by issue or reference date), lagged values, recent revision magnitudes, and short-term disease activity levels. This allows the model to capture the information that parametric delay models exploit while remaining relatively robust to misspecification and distributional shifts.

Furthermore, our framework enables timely revisions of epidemic forecasting outputs, mitigating the risk of misleading situational awareness and suboptimal decision making. Notably, Delphi-RF achieves competitive or superior forecast accuracy compared to existing methods such as NobBS and Epinowcast, while also demonstrating a 10x-100x or more improvement in computational efficiency.

Given that our method still faces challenges when epidemic disease activity levels change dramatically, particularly when encountering revision patterns not seen in historical data, a promising direction for future research is the development of a revision alerting system. This system would detect distribution shifts and predict the quality of revision forecasts based solely on early revisions, such as those available within the first one or two weeks. Such a system would complement the current framework by allowing users to proactively address potential declines in forecast accuracy and provide timely notifications to mitigate the risks associated with forecast degradation.

While our method performs robustly under typical conditions, it faces challenges during periods of dramatic shifts in epidemic activity, particularly when revision patterns diverge from those observed in historical data. A promising direction for future research is the development of a revision alerting system capable of detecting distributional shifts and providing early estimates of forecast reliability when targets are not yet available for direct evaluation. Such a system would complement the current framework by enabling users to proactively respond to potential declines in forecast accuracy and issue timely alerts to mitigate the risks associated with forecast degradation.

## Supporting information

**S1 Table. Runtime comparison across methods for daily data at different training windows.** This table presents the mean and standard error of the mean (SEM) for computing time (in seconds) required by different forecasting methods, applied to daily COVID-19 datasets. Results are reported per reference date and location. Models are trained and used to generate forecasts every 30 days for CHNG insurance claims data and every 7 days for MA-DPH confirmed case data. The comparison is performed under two training window settings (180 and 365 days), with all other configurations held constant to ensure a fair evaluation of runtime differences across methods.
(XLSX)

**S1 Appendix. Complete Set of DelphiRF Revision Count Forecasts by Reference Date.** Shown are DelphiRF count revision forecasts for CHNG outpatient insurance claims data at a reporting lag of 7 days across U.S. states. The model is re-trained every 30 days using training windows of 180 and 365 days, respectively. The figure format follows that of Figs 6 and 7.
(PDF)

**S1 Fig. Lag convergence patterns across states.** (A) Comparison of the lags required for convergence across states, shown for a sample of different reference dates based on CHNG outpatient COVID-19 insurance claims. (B) Same as in (A), but based on CHNG outpatient total insurance claims data.
(TIFF)

**S2 Fig. Temporal annotation of trend categories.**
(TIFF)

**S3 Fig. Forecast accuracy across different regularization strengths ($\lambda$).**
(TIFF)

**S4 Fig. Forecast accuracy across different decay parameter values ($\gamma$).**
(TIFF)

**S5 Fig. Forecast accuracy across different lag-padding values ($c$).**
(TIFF)

## Acknowledgments

We are deeply grateful to Ryan Tibshirani, Logan Brooks, and Will Townes for their thoughtful discussions and constructive feedback. We also acknowledge Change Healthcare and QuidelOrtho for their data partnerships and continued collaboration.

## Author contributions

**Conceptualization:** Jingjing Tang, Aaron Rumack, Bryan Wilder, Roni Rosenfeld.

**Formal analysis:** Jingjing Tang.

**Investigation:** Jingjing Tang.

**Methodology:** Jingjing Tang, Aaron Rumack.

**Software:** Jingjing Tang.

**Supervision:** Bryan Wilder, Roni Rosenfeld.

**Writing – original draft:** Jingjing Tang.

**Writing – review & editing:** Jingjing Tang, Bryan Wilder, Roni Rosenfeld.

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
