## [Decision Letter · Decision Letter 0]

25 Jun 2025

PCOMPBIOL-D-25-00938

Real-time Forecasting of Data Revisions in Epidemic Surveillance Streams

PLOS Computational Biology

Dear Dr. Tang,

Thank you for submitting your manuscript to PLOS Computational Biology. After careful consideration, we feel that it has merit but does not fully meet PLOS Computational Biology's publication criteria as it currently stands. Therefore, we invite you to submit a revised version of the manuscript that addresses the points raised during the review process.

Please submit your revised manuscript within 30 days Aug 25 2025 11:59PM. If you will need more time than this to complete your revisions, please reply to this message or contact the journal office at ploscompbiol@plos.org. Please include the following items when submitting your revised manuscript:

We look forward to receiving your revised manuscript.

Kind regards,

Philipp Martin Altrock, Ph.D.

Academic Editor

PLOS Computational Biology

Roger Kouyos

Section Editor

PLOS Computational Biology

**Journal Requirements:**

3) We have noticed that you cited "(Table 2, Appendix B)" on page 18. However, it is not included in the Supporting Information file. Please check the citation of the supplementary table and amend it accordingly.

**Reviewers' comments:**

Reviewer's Responses to Questions

**Comments to the Authors:**

**Please note that one of the reviews is uploaded as an attachment.**

Reviewer #1: Summary

This article addresses the problem of data revisions in epidemiological surveillance. This problem, often due to variable data reporting delays, can lead to errors and bias in epidemiological inference. As this problem is prevalent, this manuscript addresses an important issue in the epidemiology community. This paper introduces a method to correct epidemiological quantities for possible data revisions. The authors compare it with state of the art methods.

The problem is clearly outlined and put into a clear scientific and public health context, including a comprehensive literature review in the introduction. The notation is clearly introduced. The model, its features and hyperparameters are clearly explained, although a less technical summary of the model could improve readability. We appreciate the discussion of properties of the selected evaluation metrics. The experimental setup is clearly explained. The results are clearly stated and sufficient intuition is given to the reader, and the limitations are stated clearly. The comparison with other methods is thorough, and the novelty of Delphi-RF is made apparent.

Originality:

Although I am no expert in that exact field, the authors make a convincing case in this manuscript that the research presented here is original.

Innovation:

The approach presented here appears innovative.

High importance to researchers in the field:

The research presented here addresses an issue which is commonly regarded as highly important in the field.

Significant biological and/or methodological insight:

The research is presented with sufficient biological and methodological insight.

Rigorous methodology:

The methodology appears here rigorous, and the evaluation of the model is sufficiently thorough.

Substantial evidence for its conclusions:

The conclusions are supported with sufficient evidence, and the limitations are clearly stated.

Specific comment: Fig 7. Is titled with “Boxenplot”, which to my (vague) knowledge refers to the Seaborn implementation of the “letter value plot”. However they look and are described in the Figure caption like regular boxplots?

Reviewer #2: See attachment

**Have the authors made all data and (if applicable) computational code underlying the findings in their manuscript fully available?**

Reviewer #1: Yes

Reviewer #2: Yes

PLOS authors have the option to publish the peer review history of their article (what does this mean?). If published, this will include your full peer review and any attached files.

Reviewer #1: No

Reviewer #2: No

**Figure resubmission:**
---

## [Decision Letter · Decision Letter 1]

6 Nov 2025

Dear PhD student Tang,

We are pleased to inform you that your manuscript 'Real-time Forecasting of Data Revisions in Epidemic Surveillance Streams' has been provisionally accepted for publication in PLOS Computational Biology.

Best regards,

Philipp Martin Altrock, Ph.D.

Academic Editor

PLOS Computational Biology

Roger Kouyos

Section Editor

PLOS Computational Biology

Reviewer's Responses to Questions

**Comments to the Authors:**

Reviewer #2: Thanks for addressing my feedback. I have no further comments.

**Have the authors made all data and (if applicable) computational code underlying the findings in their manuscript fully available?**

Reviewer #2: Yes

PLOS authors have the option to publish the peer review history of their article (what does this mean?). If published, this will include your full peer review and any attached files.

Reviewer #2: No

---

## [Editor Report · Acceptance letter]

PCOMPBIOL-D-25-00938R1

Real-time Forecasting of Data Revisions in Epidemic Surveillance Streams

Dear Dr Tang,

I am pleased to inform you that your manuscript has been formally accepted for publication in PLOS Computational Biology. Your manuscript is now with our production department and you will be notified of the publication date in due course.

With kind regards,

Anita Estes
